# Bullying among nursing university students: Prevalence, characteristics, and public health implications

**Fatema Alajaimi[1], Mohammed Al-Badi[1], Hoor Alhabsi[1], Maria AL Azri[1], Shahd Al-Ghawi[1], Maryam Alwahaibi[1], Sanjay Jaju[2], Nasar Alwahaibi** [1]*

**1** Department of Biomedical Science, College of Medicine and Health Sciences, Sultan Qaboos University, Muscat, Sultanate of Oman, **2** Family Medicine & Public Health Department, College of Medicine & Health Sciences, Sultan Qaboos University, Muscat, Sultanate of Oman

\* nasar@squ.edu.om

## Abstract

Bullying among university students poses a significant public health concern, yet limited research addresses its prevalence and impact in nursing education. This study assessed the prevalence, types, effects, and contributing factors of bullying among nursing students to inform institutional and educational interventions. A cross-sectional study was conducted between October 2024 and March 2025, using convenience sampling to recruit 240 students who completed a structured questionnaire on sociodemographic characteristics and bullying experiences. Data were analyzed using descriptive statistics, Chi-square tests, and logistic regression. The prevalence of bullying was 26.3%, with verbal and emotional abuse most frequently reported, primarily by classmates in classroom settings. Reported consequences included disengagement, depression, and reduced motivation, while the majority of victims did not disclose their experiences. Early-phase students were at greater risk, and lower internet use was associated with reduced odds of bullying. These findings highlight the psychological and academic consequences of bullying. They underscore the need for institutional policies, supportive reporting mechanisms, and targeted health promotion strategies, particularly for early-phase students, to create a safer and more resilient learning environment.

## Introduction

Bullying is a recurring and detrimental behavior that involves repeated physical, verbal, psychological, or social aggression, often intended to cause harm, distress, or assert control over others [1–3]. In academic environments, especially within health professional education, it may also present through behaviors such as intimidation, exclusion, or inappropriate use of authority, impacting students during formative stages of their training [4].

**Data availability statement:** The data that support the findings of descriptive analysis of this study are available in Zenodo with the identifier given below: https://doi.org/10.5281/zenodo.15735213.

**Funding:** The authors received no specific funding for this work.

**Competing interests:** The authors have declared that no competing interests exist.

While the broader educational literature has acknowledged the harmful impact of bullying, most research in health education has focused on practicing clinicians, leaving the experiences of healthcare students insufficiently examined [5,6]. Evidence indicates that bullying in nursing education is associated with heightened stress, reduced self-esteem, academic difficulties, and in some cases, attrition from nursing programs [7–10]. Structural factors such as ranked clinical environments, high-stakes placements, and repeated exposure to authority figures increase the likelihood of such negative experiences [11]. From a public health education standpoint, these findings underscore the need for systemic strategies to foster resilience, well-being, and professional identity formation in nursing students [12,13].

Nursing students are not only learners but also emerging public health advocates and frontline caregivers. As they acquire clinical competencies, they are also expected to embody values such as empathy, communication, and ethical decision-making [14,15]. Through extensive academic and hands-on training, they gain direct exposure to real-world clinical settings and interprofessional dynamics, which are essential for their development into competent practitioners [16,17]. Supporting their well-being throughout this journey is essential, not only for educational success but also for sustaining a resilient and ethical nursing workforce capable of addressing global health challenges [18].

Despite growing global recognition of bullying in academic settings, data specific to nursing students in many regions remains scarce [19]. To the best of our knowledge, in Oman, no published studies to date have comprehensively assessed bullying among nursing students across both academic and clinical environments. This study aimed to fill that gap by examining the prevalence, characteristics, and effects of bullying in a university-based nursing program, thereby contributing to health promotion efforts in higher education and supporting the creation of safe and inclusive learning environments.

## Methods

### Ethical considerations

This study was conducted in accordance with the ethical principles outlined in the Declaration of Helsinki and received approval from the Medical Research Ethics Committee (MREC) at the College of Medicine and Health Sciences, Sultan Qaboos University (SQU), Oman (Approval No. MREC #3300). Prior to participation, all students were given detailed information about the purpose of the study, their rights as participants, and the confidentiality measures in place. Written informed consent was obtained from all participants, including those aged 17 years, with additional consent provided by their legal guardians. Participation was entirely voluntary, and students were assured that they could withdraw from the study at any stage without any consequences. Ethical safeguards were rigorously upheld throughout the research process to protect the participants' rights and well-being.

### Study design

This was a cross-sectional observational study conducted from October 07, 2024 to March 31, 2025. A convenience sampling method was used to recruit participants

from the College of Nursing at SQU. The study targeted all currently enrolled nursing students, regardless of age or academic year. Students outside the nursing program or those who chose not to participate were excluded. The survey aimed to capture a snapshot of students' experiences and perceptions related to bullying during their academic journey.

## Participants

Participants were nursing students enrolled in the undergraduate program at Sultan Qaboos University, a leading public university located in Al-Seeb, Oman. Admission into the nursing program is based on high school academic performance. Students either enter directly into the four-year program or, if required, complete a one-year foundation program covering English language, mathematics, information technology, and study skills. For analysis purposes, students were grouped into an early phase (foundation year through year 2) and a late phase (years 3–5).

## Sample size calculation

The sample size was determined using the formula for finite population sampling: $n = NZ^2p(1-p)/[d^2(N-1) + Z^2p(1-p)]$, where $n$ is the sample size, $N$ is the total number of nursing students (500), $Z$ is the standard normal deviate for a 95% confidence level (1.96), $p$ is the assumed population proportion (0.5), and $d$ is the margin of error (0.05) [20]. To account for potential non-responses or improperly completed surveys, an additional 10% was added to the calculated figure, resulting in a final target sample size of 240 students.

## Pilot testing and reliability

A pilot study was carried out with 15 nursing students to assess the clarity, relevance, and reliability of the questionnaire. Feedback from the pilot group led to minor revisions to enhance comprehension and flow. The internal consistency of the questionnaire was evaluated using Cronbach's alpha, which yielded a value of 0.68. While this falls slightly below the conventional threshold of 0.70, values between 0.60 and 0.70 are generally regarded as acceptable in exploratory research, particularly when instruments include diverse items or are adapted to different cultural contexts [21,22]. This result therefore indicates moderate reliability and highlights the need for future studies to refine and further validate the instrument.

## Data collection

Data were collected using an online, self-administered questionnaire distributed via Google Forms. The survey was shared with students through institutional emails and social media platforms such as WhatsApp to maximize accessibility and participation. The questionnaire was available in both English and Arabic, and translations were reviewed by native speakers to ensure accuracy. The anonymity of responses was maintained, and standardized instructions were provided to reduce potential biases. To ensure data quality, responses were checked, and incomplete or inconsistent entries were excluded during data cleaning. Access to the questionnaire was restricted to nursing students, with participants required to confirm their enrollment status at the start of the survey.

## Questionnaire structure

This study used a researcher-developed questionnaire that included a combination of original items and questions adapted from previously validated instruments in the literature [23–25]. The questionnaire was organized into four sections. The first section introduced the study, outlining its purpose, significance, ethical approval, and operational details. The second section addressed informed consent, requiring participants to confirm their agreement before proceeding. The third section collected sociodemographic data including age, gender, marital status, academic year, parental education level, household income, general health status, and internet usage. The final section explored bullying-related experiences, focusing on frequency, types, perpetrators, reasons for bullying, and its perceived effects on academic and

emotional well-being. Content validity was assessed by a panel of academic and clinical experts, who provided feedback to ensure the questions adequately addressed the topic of bullying among nursing students. Periodic reminders were also delivered through official institutional emails and verified College of Nursing WhatsApp groups throughout the data collection phase.

## Data analysis

Data were analyzed using IBM SPSS Statistics version 29.0 (IBM Corp., Armonk, NY, USA). Categorical variables were summarized using frequencies and percentages to describe the characteristics of the study population. Associations between categorical variables and bullying status were examined using the Chi-square test, which is appropriate for detecting differences between groups with categorical data. To identify independent predictors of bullying, binary logistic regression was conducted since the outcome variable (being bullied vs. not bullied) was dichotomous. Variables with a crude p-value < 0.25 in the bivariate analysis were included in the multivariate model to ensure that potential confounders were not overlooked [26]. Statistical significance was determined using a p-value < 0.05.

## Results

S1 Table provides an overview of the demographic characteristics of the 240 nursing students who participated in the study. Most participants were aged 20–22 years (44.2%) and the majority were females (74.2%). Over half lived with roommates or in shared accommodation (54.2%), while a large proportion were single (92.1%) and in the later phase of their studies (60.1%). The majority of students came from non-urban areas (57.5%). In terms of parental education, most fathers (68.8%) and mothers (84.2%) had not completed university-level education. The majority of students reported a middle-income family background (60.8%) and rated their health 12.1% "good," 47.5% "very good," and 37.1% "excellent." Nearly all students (97.9%) reported using the internet multiple times a day, primarily accessing it via Wi-Fi at the university (94.2%) and at home (83.8%).

Table 1 provides an overview of the prevalence and patterns of bullying among the participating students. A total of 26.3% (95% C.I.: 20.8% - 32.3%) of students experienced bullying during their studies at SQU, and 42.9% had witnessed bullying events. Nearly 43.3% had been bullied at school, while 32.1% reported being bullied at home (Fig 1). Among those who witnessed bullying, two-thirds (67%) tried to interfere. A smaller proportion (12.1%) admitted that they bullied others, with face-to-face messaging being the most common method (8.3%). The main reason reported for bullying others was "for fun" (9.6%). Notably, a large majority (89.6%) agreed that there should be a law at SQU to protect students from bullying.

S2 Table presents the detailed characteristics and consequences of bullying among 63 students who reported being bullied at SQU. Over half (54.0%) were bullied a few times in their academic term, and 93.7% were bullied by SQU mates. Verbal bullying (98.4%) and emotional/mental abuse (36.5%) were the most frequently reported types (Fig 2). The most common negative effects among these bullied students were disengagement (42.9%), depression (33.3%), and lack of motivation (31.7%). For the mean of bullying, most (82.5%) were through face-to-face messaging, and 42.9% through friends. Most occurred in classrooms (73.0%), corridors (44.4%), and resting rooms (41.3%). Bullying was made due to personal traits (58.7%), personal differences (50.8%), and academic major (46.0%). Majority of them (93.7%) did not report the incidents. In this regard, 74.6% felt that it was not important, and 30.2% hoped it will stop itself.

Table 2 summarizes the crude associations between student characteristics and the likelihood of being bullied at SQU. In this regard, among those who had been bullied at SQU, 74.6% had a history of school bullying, 60.3% had been bullied at home, and 63.5% had witnessed bullying at the university compared 32.2%, 22.0%, and 35.6% among non-bullied students respectively. These differences were statistically significant (p < 0.001). Among bullied students, 22.2% reported having bullied others, compared to only 8.5% of those who had not been bullied (p = 0.004). Gender also showed a significant difference. In this regard, male students were more prevalent among the bullied students compared to non-bullied (38.1% vs. 21.5% respectively,

**Table 1. Distribution of bullying experiences and related factors among nursing university students.**

| Variables | n (240) | % |
|---|---|---|
| **Have you ever been bullied at school? (n=240)** | | |
| Yes | 104 | 43.3 |
| No | 136 | 56.7 |
| **Have you ever been bullied at home? (n=240)** | | |
| Yes | 77 | 32.1 |
| No | 163 | 67.9 |
| **Have you ever experienced bullying at SQU during your study? (n=240)** | | |
| Yes | 63 | 26.3 |
| No | 177 | 73.8 |
| **Have you witnessed bullying at SQU? (n=240)** | | |
| Yes | 103 | 42.9 |
| No | 137 | 57.1 |
| *If yes, what was your reaction? (n=103)* | | |
| Tried to interfere | 69 | 67 |
| Not reacted in any way | 34 | 33 |
| **Have you ever bullied other students? (n=240)** | | |
| Yes | 29 | 12.1 |
| No | 211 | 87.9 |
| **Have you used any of the following to bully other students? (n=29)** | | |
| Online video clips of them | 7 | 2.9 |
| Chatroom | 7 | 2.9 |
| Through friends | 12 | 5 |
| Face to face messaging | 20 | 8.3 |
| Picture messages | 8 | 3.3 |
| **What makes you bully other students? (n=29)** | | |
| Due to personal issues | 7 | 2.9 |
| For fun | 23 | 9.6 |
| To feel more powerful | 6 | 2.5 |
| **Do you think there should be a law at SQU to protect students from bullying? (n=240)** | | |
| Yes | 215 | 89.6 |
| No | 25 | 10.4 |

p=0.010). Among the bullied, the late phase students were reported more frequent (75.8%) compared to non-bullied students (54.5%), which was statistically significant (p=0.003). Place of origin was also associated with bullying as more bullied students came from urban areas (54.0%) compared to 38.4% among non-bullied students (p=0.032). Use of Wi-Fi hotspots from friends or family was significantly more common among bullied students (39.7%) than non-bullied (25.4%), with a p-value of 0.032. However, other factors including age, marital status, residence, parents' education, income level, general health, and most internet usage variables, did not show significant differences between the compared groups (p>0.05) (Table 2).

Table 3 presents the adjusted logistic regression model identifying independent predictors of bullying. A history of bullying at school had lower odds of being bullied at SQU (OR=0.222, p<0.001), and those who were bullied at home had also lower odds of being bullied at SQU (OR=0.262, p<0.001). Students in the early phase of their studies were over 5 times more likely to be bullied at SQU compared to those in the late phase (OR=5.29, p=0.030). Students who used the internet once a day were significantly less likely to be bullied (OR=0.059, p=0.012). Other factors, such as witnessed bullying at

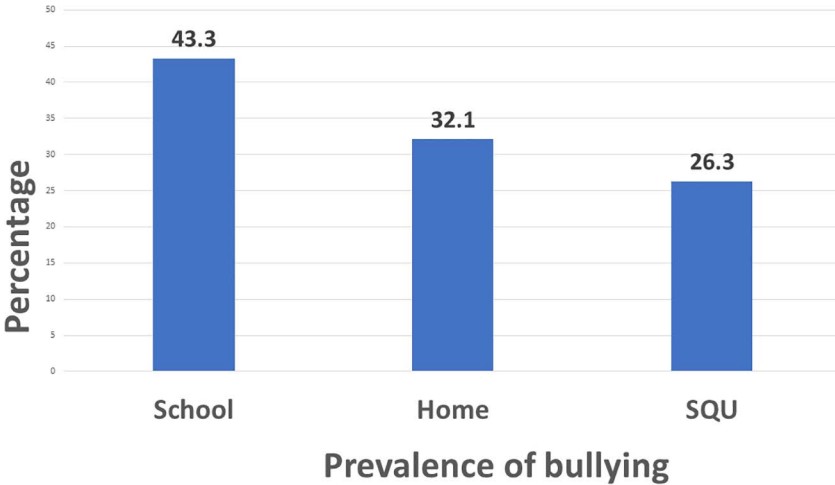

**Fig 1. The prevalence of bullying at school, home and Sultan Qaboos University (SQU) among nursing university students.**

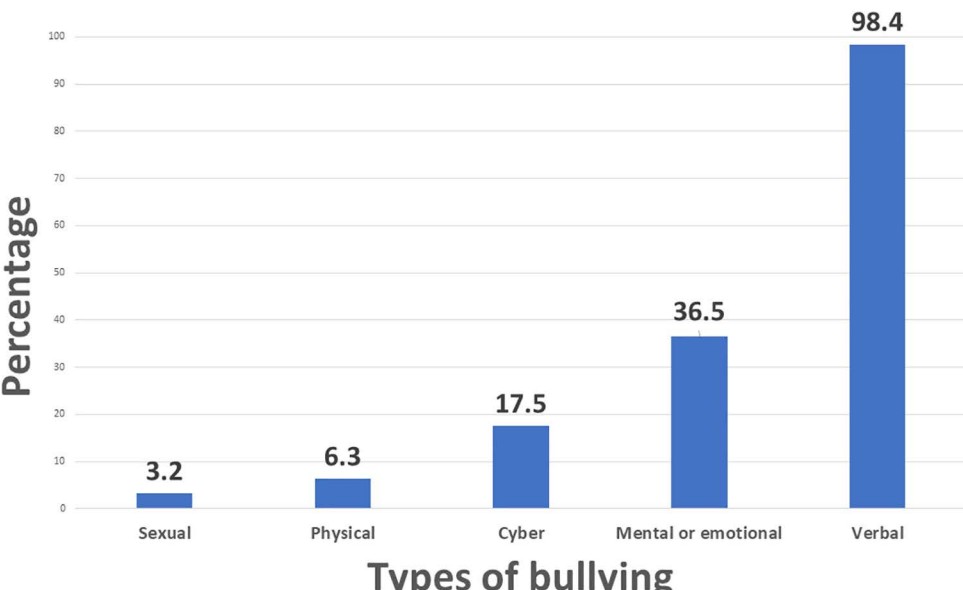

**Fig 2. Types of bullying experienced by nursing university students.**

SQU, having bullied others, gender, age, living arrangements, place of origin, and types of internet access did not show any significant association in the adjusted model (p > 0.05).

## Discussion

The aim of this study was to explore the prevalence, forms, and impact of bullying among nursing students at a university. The current study found a bullying prevalence rate of 26.3% among nursing students, which is within the lower to moderate range compared to global figures reported in the literature. Some studies have documented much higher rates, reaching up to 96% [27], while others have reported lower incidences ranging from 0.5% to 9% depending on how bullying

**Table 2. Comparison of sociodemographic and behavioral factors between bullied and non-bullied nursing university students.**

| Variables | Have you ever experienced bullying at SQU during your study? | | p-Value* |
|---|---|---|---|
| | Yes (n=63) | No (n=177) | |
| | n (%) | n (%) | |
| **Have you ever been bullied at school? (n=240)** | | | |
| Yes | 47 (74.6) | 57 (32.2) | <0.001 |
| No | 16 (25.4) | 120 (67.8) | |
| **Have you ever been bullied at home? (n=240)** | | | |
| Yes | 38 (60.3) | 39 (22.0) | <0.001 |
| No | 25 (39.7) | 138 (78.0) | |
| **Have you witnessed bullying at SQU? (n=240)** | | | |
| Yes | 40 (63.5) | 63 (35.6) | <0.001 |
| No | 23 (36.5) | 114 (64.4) | |
| **Have you ever bullied other students? (n=240)** | | | |
| Yes | 14 (22.2) | 15 (8.5) | 0.004 |
| No | 49 (77.8) | 162 (91.5) | |
| **Do you think there should be a law at SQU to protect students from bullying? (n=240)** | | | |
| Yes | 55 (87.3) | 160 (90.4) | 0.49 |
| No | 8 (12.7) | 17 (9.6) | |
| **Age (n=240)** | | | |
| 17-19 | 12 (19.0) | 56 (31.6) | 0.197 |
| 20-22 | 29 (46.0) | 77 (43.5) | |
| 23-25 | 17 (27.0) | 29 (16.4) | |
| ≥26 | 5 (7.9) | 15 (8.5) | |
| **Gender (n=240)** | | | |
| Male | 24 (38.1) | 38 (21.5) | 0.01 |
| Female | 39 (61.9) | 139 (78.5) | |
| **Where do you live? (n=240)** | | | |
| Alone | 8 (12.7) | 11 (6.2) | 0.24 |
| With roommates/ In a flat | 31 (49.2) | 99 (55.9) | |
| With your own family | 24 (38.1) | 67 (37.9) | |
| **Marital status (n=239)** | | | |
| Single | 58 (92.1) | 162 (92.0) | 0.996 |
| Married | 5 (7.9) | 14 (8.0) | |
| **Study phase (n=238)** | | | |
| Early | 15 (24.2) | 80 (45.5) | 0.003 |
| Late | 47 (75.8) | 96 (54.5) | |
| **Place of original residence (n=240)** | | | |
| Urban | 34 (54.0) | 68 (38.4) | 0.032 |
| Non-Urban | 29 (46.0) | 109 (61.6) | |
| **Father's educational level (n=240)** | | | |
| Pre-University education | 41 (65.1) | 124 (70.1) | 0.752 |
| Undergraduate | 15 (23.8) | 35 (19.8) | |
| Postgraduate | 7 (11.1) | 18 (10.2) | |
| **Mother's educational level (n=240)** | | | |
| Pre-University education | 50 (79.4) | 152 (85.9) | 0.464 |
| Undergraduate | 10 (15.9) | 20 (11.3) | |

*(Continued)*

**Table 2.** (Continued)

| Variables | Have you ever experienced bullying at SQU during your study? | | p -Value* |
|---|---|---|---|
| | **Yes (n = 63)** **n (%)** | **No (n = 177)** **n (%)** | |
| Postgraduate | 3 (4.8) | 5 (2.8) | |
| **Family economic status – monthly income in OMR (n = 240)** | | | |
| Low (<500) | 12 (19.0) | 35 (19.8) | 0.391 |
| Middle (500–1500) | 35 (55.6) | 111 (62.7) | |
| High (>1500) | 16 (25.4) | 31 (17.5) | |
| **How would you rate your general health? (n = 240)** | | | |
| Poor/ Fair | 1 (1.6) | 7 (4.0) | 0.449 |
| Good | 10 (15.9) | 19 (10.7) | |
| Very good | 32 (50.8) | 82 (46.3) | |
| Excellent | 20 (31.7) | 69 (39.0) | |
| **How often do you use the internet? (n = 240)** | | | |
| Once a day/others | 3 (4.8) | 2 (1.1) | 0.115 |
| Multiple Times a Day | 60 (95.2) | 175 (98.9) | |
| **Using Wi-Fi connection at SQU campus (n = 240)** | | | |
| Yes | 61 (96.8) | 165 (93.2) | 0.367 |
| No | 2 (3.2) | 12 (6.8) | |
| **Using Wi-Fi connection at Home (n = 240)** | | | |
| Yes | 55 (87.3) | 146 (82.5) | 0.374 |
| No | 8 (12.7) | 31 (17.5) | |
| **Using personal mobile data (n = 240)** | | | |
| Yes | 54 (85.7) | 133 (75.1) | 0.082 |
| No | 9 (14.3) | 44 (24.9) | |
| **Wi-Fi hotspots from Friends/Family (n = 240)** | | | |
| Yes | 25 (39.7) | 45 (25.4) | 0.032 |
| No | 38 (60.3) | 132 (74.6) | |
| **Wi-Fi hotspots in public places (n = 240)** | | | |
| Yes | 16 (25.4) | 26 (14.7) | 0.055 |
| No | 47 (74.6) | 151 (85.3) | |

was defined and measured [28]. This variation may stem from differences in study design, cultural perceptions of bullying, and methodological tools used [1].

Nursing students who had experienced bullying at school or at home were significantly less likely to be bullied, as indicated by odds ratios of 0.222 and 0.262, respectively, both with p-values < 0.001. These findings suggest a protective association, which may appear contradictory given the well-documented negative impacts of bullying. One possible explanation is that students previously exposed to bullying may have developed greater resilience, coping strategies, or awareness that reduce their vulnerability to similar experiences in new environments. Alternatively, these results may reflect differences in perception, reporting behavior, or the influence of unmeasured confounding factors such as current social support or psychological interventions. These unexpected results show that more research is needed to better understand how past bullying experiences affect students later in life.

The prevalence of bullying among nursing United Kingdom students and Australian were 35% and 50%, respectively [29]. Similar findings were also reported with slightly different ratios as with Italian nursing students 34% [30],

**Table 3. Logistic regression analysis of factors associated with bullying among nursing university students (n = 240).**

| Variables | B coefficient | p-Value | Odds | 95% C.I. for Odds | |
|---|---|---|---|---|---|
| | | | | Lower | Upper |
| Have you ever been bullied at school? (Ref: No) | | | | | |
| Yes | -1.506 | <0.001 | 0.222 | 0.1 | 0.491 |
| Have you ever been bullied at home? (Ref: No) | | | | | |
| Yes | -1.341 | <0.001 | 0.262 | 0.121 | 0.564 |
| Have you witnessed bullying at SQU? (Ref: No) | | | | | |
| Yes | -0.619 | 0.096 | 0.538 | 0.26 | 1.117 |
| Have you ever bullied other students? (Ref: No) | | | | | |
| Yes | -0.65 | 0.217 | 0.522 | 0.186 | 1.466 |
| Age groups (Ref: ≥ 26) | | | | | |
| 17-19 | 0.029 | 0.975 | 1.029 | 0.169 | 6.267 |
| 20-22 | 1.029 | 0.182 | 2.798 | 0.616 | 12.7 |
| 23-25 | 0.847 | 0.301 | 2.333 | 0.468 | 11.634 |
| Gender (Ref: Female) | | | | | |
| Male | -0.528 | 0.228 | 0.59 | 0.25 | 1.392 |
| Where do you live? (Ref: With own family) | | | | | |
| Alone | -0.707 | 0.272 | 0.493 | 0.14 | 1.742 |
| With roommates/ In a flat | -0.554 | 0.209 | 0.574 | 0.242 | 1.364 |
| Year of study (Ref: Late phase) | | | | | |
| Early phase | 1.665 | 0.03 | 5.288 | 1.178 | 23.745 |
| Place of original residence (Ref: Non-Urban) | | | | | |
| Urban | -0.437 | 0.244 | 0.646 | 0.31 | 1.348 |
| How often do you use the internet? (Ref: Multiple Times a Day) | | | | | |
| Once a day/others | -2.836 | 0.012 | 0.059 | 0.006 | 0.534 |
| Personal mobile data (Ref: No) | | | | | |
| Yes | -0.507 | 0.326 | 0.602 | 0.219 | 1.657 |
| Wi-Fi hotspots from Friends/Family (Ref: No) | | | | | |
| Yes | -0.489 | 0.336 | 0.613 | 0.226 | 1.662 |
| Wi-Fi hotspots in public places (Ref: No) | | | | | |
| Yes | 0.006 | 0.993 | 1.006 | 0.311 | 3.255 |

and with the Turkish nursing students was 60% [31]. Another study conducted in Egypt found that 51.9% of the participating student nurses had experienced bullying [32]. In China, two studies represented almost the same findings with 58.17% and 55.75% reported experiencing bullying during their nursing study [33,34].

Verbal bullying was the most prevalent type reported in this study (98.4%), followed by mental or emotional bullying (36.5%). These findings are consistent with earlier research, where verbal abuse, such as ridicule, negative comments, or public humiliation, was cited as the most common form of bullying among nursing students during their study [35,36]. The present study found that sexual (3.2%) and physical (6.3%) bullying was low among the nursing students. Several studies reported similar findings [4,27,37]. Although physical and sexual forms of bullying are less frequently reported, the psychological impact of verbal and emotional bullying is profound, often resulting in long-term emotional and academic consequences [38,39].

Our data also highlighted those classmates were the primary perpetrators (93.7%), followed by teachers and instructors (42.9%). This aligns with studies emphasizing horizontal violence, where peers at similar hierarchical levels engage in

bullying [29,40]. Other two studies in Saudi Arabia reported similar findings [13,41]. However, other recent study among nursing students in clinical placement in China showed that the most common perpetrators were patients (31.05%), registered nurses (30.39%), and patients' family and friends (26.80%) [34]. Similar findings were also reported [29]. Although much attention has traditionally been placed on bullying by clinical nurses or superiors, evidence suggests that peer-related aggression is equally, if not more, damaging, due to the breakdown of collegial support [42,43].

The reported consequences of bullying in our study, include disengagement (42.9%), depression (33.3%), lack of motivation (31.7%), and feeling low (31.7%), which reflects the serious psychological effects takes on students. These outcomes are consistent with those identified in prior research, which emphasize emotional distress, loss of self-confidence, and fear as typical consequences of bullying [13,44]. Early exposure to such toxic environments can disrupt students' development of a professional identity and hinder their preparedness for clinical practice [45]. Other studies reported anxiety, panic attacks, and even withdrawal from nursing education [27,38,46]. These impacts not only threaten student well-being but also compromise patient safety and the quality of future healthcare delivery [47,48].

In terms of bullying methods, face-to-face interactions were the dominant channel (82.5%), confirming the findings of other study, which noted that most bullying incidents occur during direct interactions in educational or clinical settings [38]. The classroom (73%) was identified as the most common location, followed by corridors (44.4%) and resting rooms (41.3%). These settings, often seen as "safe zones" for learning, become sources of distress when bullying takes place [6,49].

When asked about perceived reasons for being targeted, students most often cited their appearance or personal traits (58.7%), personal differences (50.8%), and their academic major (46%). This subjective attribution resonates with previous qualitative studies where students internalized bullying as related to their identity or perceived social standing [1,50,51].

Notably, the majority (93.7%) of students chose not to report their experiences of bullying. This is consistent with international studies showing non-reporting rates as high as 89–94% [5,44]. The main reasons were the belief that the issue was not important (74.6%) or hope that it would stop on its own (30.3%). These findings are closely aligned with themes in the systematic review, which identified fear of retaliation, lack of confidence in reporting systems, and a perception of reporting as ineffective as key deterrents to formal complaints [52,53].

A recent study emphasized that inadequate coping resources significantly correlate with higher bullying exposure [33]. Although our study did not measure coping resources directly, the low reporting rates and emotional impacts observed suggest insufficient institutional support and personal strategies to address bullying. This underscores the need for educational institutions to prioritize resilience-building, clear anti-bullying policies, and supportive reporting systems. This study underscores the urgent need for nursing institutions to implement clear, accessible reporting mechanisms and anti-bullying policies. Interventions such as faculty training, peer support groups, and student empowerment programs have shown promising results in reducing the incidence and impact of bullying [54,55]. Moreover, early education on recognizing and addressing bullying can play a key role in breaking the cycle of abuse perpetuated from the academic setting into the clinical workplace [56].

The logistic regression analysis revealed several key factors associated with bullying among nursing university students. Students in the early phase of their studies were significantly more likely to report being bullied compared to those in the later phase, with an odds ratio of 5.288 (p = 0.030), suggesting they were over five times more likely to experience bullying. This increased vulnerability may be due to their limited familiarity with the university environment, academic expectations, and social dynamics, as well as underdeveloped coping mechanisms and support networks. Nearly all students reported frequent internet access, primarily through Wi-Fi at the university and at home. Notably, those who used the internet once a day or less were significantly less likely to report being bullied compared to frequent users (OR = 0.059, p = 0.012). This suggests that high levels of online activity may increase exposure to bullying behaviors, particularly through digital platforms, even though most reported bullying occurred face-to-face. The internet may therefore act as

both a medium for cyberbullying and a facilitator of negative peer interactions. In light of these findings, universities should consider integrating digital literacy and online safety training into student orientation programs, particularly for early-phase students. Strengthening peer mentoring, resilience-building workshops, and awareness campaigns on responsible internet use could help reduce bullying risks and provide a safer, more supportive academic environment. These findings underscore the importance of addressing both traditional and online forms of bullying when developing preventive strategies and health promotion programs in nursing education.

## Limitations of the study

This study has several limitations. First, it was conducted at a single governmental university, which may limit generalizability, and the findings may not fully represent all nursing students in Oman or the wider Gulf region. Second, the use of convenience sampling restricts representativeness, and selection bias cannot be excluded, as students more affected by bullying may have been more motivated to participate. Third, reliance on self-reported data may have introduced recall or reporting bias. Fourth, the pilot questionnaire showed moderate internal consistency (Cronbach's alpha $= 0.68$), which is acceptable in exploratory research but warrants caution and further refinement in future studies. Fifth, pilot testing was conducted with nursing students, the same population as the main study, which may have introduced some bias; ideally, it would involve a similar but non-identical group. Finally, the cross-sectional design prevents causal inferences about the long-term consequences of bullying.

## Conclusion

This study underscores bullying as a pressing public health concern within nursing education, with verbal and emotional abuse frequently reported and linked to adverse psychological outcomes such as disengagement, depression, and reduced motivation. The findings reveal critical risk factors, including early-phase enrollment and high internet use, while prior experiences of bullying may foster resilience. The high prevalence of non-reporting highlights the urgent need for culturally sensitive health education strategies, structured reporting systems, and targeted interventions. Promoting safe, inclusive, and empowering academic environments is essential to advancing student well-being and strengthening the future healthcare workforce.

## Supporting information

**S1 Table. Sociodemographic characteristics of nursing university students.**
(DOCX)

**S2 Table. Bullying experiences and related factors among nursing university students who reported being bullied.**
(DOCX)

## Acknowledgments

The authors would like to thank all nursing students at the College of Nursing, Sultan Qaboos University, for their participation in this study.

## Author contributions

**Conceptualization:** Nasar Alwahaibi.

**Data curation:** Fatema Alajaimi, Mohammed Al-Badi, Hoor Alhabsi, Mariya AL Azri, Shahd Al-Ghawi, Maryam Alwahaibi, Sanjay Jaju.

**Formal analysis:** Fatema Alajaimi, Mohammed Al-Badi, Hoor Alhabsi, Mariya AL Azri, Shahd Al-Ghawi, Maryam Alwahaibi, Sanjay Jaju.

**Investigation:** Fatema Alajaimi, Mohammed Al-Badi, Hoor Alhabsi, Mariya AL Azri, Shahd Al-Ghawi, Maryam Alwahaibi, Sanjay Jaju.

**Methodology:** Fatema Alajaimi, Mohammed Al-Badi, Hoor Alhabsi, Mariya AL Azri, Shahd Al-Ghawi, Maryam Alwahaibi, Sanjay Jaju.

**Project administration:** Nasar Alwahaibi.

**Resources:** Nasar Alwahaibi.

**Supervision:** Nasar Alwahaibi.

**Validation:** Fatema Alajaimi, Mohammed Al-Badi, Hoor Alhabsi, Mariya AL Azri, Shahd Al-Ghawi, Maryam Alwahaibi, Sanjay Jaju.

**Writing – original draft:** Nasar Alwahaibi.

**Writing – review & editing:** Fatema Alajaimi, Mohammed Al-Badi, Hoor Alhabsi, Mariya AL Azri, Shahd Al-Ghawi, Maryam Alwahaibi, Sanjay Jaju, Nasar Alwahaibi.

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
