## [Decision Letter · Decision Letter 0]

29 Sep 2025

PGPH-D-25-01719

Bullying among nursing university students: prevalence, characteristics, and public health implications

Dear Dr. Alwahaibi,

Thank you for submitting your manuscript to PLOS Global Public Health. After careful consideration, we feel that it has merit but does not fully meet PLOS Global Public Health’s publication criteria as it currently stands. Therefore, we invite you to submit a revised version of the manuscript that addresses the points raised during the review process.

We look forward to receiving your revised manuscript.

Kind regards,

Somayeh Hessam

Academic Editor

Journal Requirements:

1. We have noticed that you have uploaded Supporting Information files, but you have not included a list of legends. Please add a full list of legends for your Supporting Information files after the references list.

Additional Editor Comments (if provided):

Reviewers' comments:

Reviewer's Responses to Questions

**Comments to the Author**

1. Does this manuscript meet PLOS Global Public Health’s publication criteria ? Is the manuscript technically sound, and do the data support the conclusions? The manuscript must describe methodologically and ethically rigorous research with conclusions that are appropriately drawn based on the data presented.

Reviewer #1: Yes

Reviewer #2: Yes

2. Has the statistical analysis been performed appropriately and rigorously?

Reviewer #1: Yes

Reviewer #2: Yes

3. Have the authors made all data underlying the findings in their manuscript fully available (please refer to the Data Availability Statement at the start of the manuscript PDF file)?

Reviewer #1: Yes

Reviewer #2: Yes

4. Is the manuscript presented in an intelligible fashion and written in standard English?

Reviewer #1: Yes

Reviewer #2: Yes

5. Review Comments to the Author

Reviewer #1: The study addresses an important issue surrounding bullying in nursing students. The topic is important and timely, but the author needs to revise the manuscript to clarify content and increase methodological rigor, and avoid disjointedness with the review of the literature, method, results, and finally conclusion. The author should address the comments to enhance the quality and contribution of the study.

Reviewer #2: The article is well-written. However, several methodological and reporting issues need clarification and strengthening to make the article suitable for publication.

1. Sampling method:

The study used convenience sampling. This limits generalizability. The authors should justify why random or stratified methods were not possible.Possible selection bias (students more affected by bullying may have been more motivated to participate).

2.Generalizability:

The study is conducted at a single university. Authors should emphasize that the results may not represent all nursing students in Oman or the Gulf region.

3.Measurement reliability:

Cronbach's alpha for the pilot questionnaire was 0.68, which is below the conventional 0.7 threshold. The authors acknowledge this, but they should discuss implications for the validity of findings.

4.Abstract: Overly detailed. Could be shortened by focusing on the most critical results and implications.

5.Tables: Some tables contain large amounts of data that may be better presented in supplementary files

6. PLOS authors have the option to publish the peer review history of their article (what does this mean? ). If published, this will include your full peer review and any attached files.

**Do you want your identity to be public for this peer review?** For information about this choice, including consent withdrawal, please see our Privacy Policy .

Reviewer #1: No

Reviewer #2: No

Figure Resubmissions:

---

## [Decision Letter · Decision Letter 1]

9 Nov 2025

PGPH-D-25-01719R1

Bullying among nursing university students: prevalence, characteristics, and public health implications

Dear Dr. Alwahaibi,

Thank you for submitting your manuscript to PLOS Global Public Health. After careful consideration, we feel that it has merit but does not fully meet PLOS Global Public Health’s publication criteria as it currently stands. Therefore, we invite you to submit a revised version of the manuscript that addresses the points raised during the review process.

We look forward to receiving your revised manuscript.

Kind regards,

Somayeh Hessam

Academic Editor

Journal Requirements:

Additional Editor Comments (if provided):

Reviewers' comments:

Reviewer's Responses to Questions

**Comments to the Author**

1. If the authors have adequately addressed your comments raised in a previous round of review and you feel that this manuscript is now acceptable for publication, you may indicate that here to bypass the “Comments to the Author” section, enter your conflict of interest statement in the “Confidential to Editor” section, and submit your "Accept" recommendation.

Reviewer #1: All comments have been addressed

Reviewer #2: All comments have been addressed

2. Does this manuscript meet PLOS Global Public Health’s publication criteria ? Is the manuscript technically sound, and do the data support the conclusions? The manuscript must describe methodologically and ethically rigorous research with conclusions that are appropriately drawn based on the data presented.

Reviewer #1: Yes

Reviewer #2: Yes

3. Has the statistical analysis been performed appropriately and rigorously?

Reviewer #1: Yes

Reviewer #2: Yes

4. Have the authors made all data underlying the findings in their manuscript fully available (please refer to the Data Availability Statement at the start of the manuscript PDF file)?

Reviewer #1: Yes

Reviewer #2: Yes

5. Is the manuscript presented in an intelligible fashion and written in standard English?

Reviewer #1: Yes

Reviewer #2: Yes

6. Review Comments to the Author

Reviewer #1: The updated manuscript responds to all comments and suggestions made by the reviewers. Changes are made and specified clearly the rebuttal letter and in the main text, to clarify the methodology, increase coherence, and to strengthen the analysis.

Reviewer #2: The study design is appropriate for its exploratory aims, the statistical analyses (descriptive statistics, chi-square tests, and multivariate logistic regression) are sound, and the conclusions are supported by the data.

1-The abstract is now more focused, but the sentence on data analysis could be tightened for brevity

2-Methods - Data Analysis (lines 165-174): The use of a crude p-value <0.25 for variable inclusion in multivariate models is justified to avoid overlooking confounders, but cite a reference for this approach (e.g., Hosmer and Lemeshow’s text on logistic regression) to support it.

3. Results (lines 200-209): The detailed breakdown in S2 Table is useful, but in the text, clarify that the percentages for bullying effects (e.g., disengagement 42.9%) are among the 63 bullied students. This avoids any misinterpretation.

4-The integration of logistic regression findings with public health implications is strong. However, suggest practical recommendations, such as digital literacy training for early-phase students to mitigate internet-related risks.

7. PLOS authors have the option to publish the peer review history of their article (what does this mean? ). If published, this will include your full peer review and any attached files.

**Do you want your identity to be public for this peer review?** For information about this choice, including consent withdrawal, please see our Privacy Policy .

Reviewer #1: **Yes: ** Jean Rose A. Abejo, RN, PhD

Reviewer #2: No

 Figure Resubmissions:

---

## [Decision Letter · Decision Letter 2]

10 Dec 2025

PGPH-D-25-01719R2

Bullying among nursing university students: prevalence, characteristics, and public health implications

Dear Dr. Alwahaibi,

Thank you for submitting your manuscript to PLOS Global Public Health. After careful consideration, we feel that it has merit but does not fully meet PLOS Global Public Health’s publication criteria as it currently stands. Therefore, we invite you to submit a revised version of the manuscript that addresses the points raised during the review process.

We look forward to receiving your revised manuscript.

Kind regards,

Somayeh Hessam

Academic Editor

Journal Requirements:

Reviewers' comments:

Reviewer's Responses to Questions

**Comments to the Author**

1. If the authors have adequately addressed your comments raised in a previous round of review and you feel that this manuscript is now acceptable for publication, you may indicate that here to bypass the “Comments to the Author” section, enter your conflict of interest statement in the “Confidential to Editor” section, and submit your "Accept" recommendation.

Reviewer #1: All comments have been addressed

Reviewer #2: All comments have been addressed

2. Does this manuscript meet PLOS Global Public Health’s publication criteria ? Is the manuscript technically sound, and do the data support the conclusions? The manuscript must describe methodologically and ethically rigorous research with conclusions that are appropriately drawn based on the data presented.

Reviewer #1: Yes

Reviewer #2: Yes

3. Has the statistical analysis been performed appropriately and rigorously?

Reviewer #1: Yes

Reviewer #2: Yes

4. Have the authors made all data underlying the findings in their manuscript fully available (please refer to the Data Availability Statement at the start of the manuscript PDF file)?

Reviewer #1: Yes

Reviewer #2: Yes

5. Is the manuscript presented in an intelligible fashion and written in standard English?

Reviewer #1: Yes

Reviewer #2: Yes

6. Review Comments to the Author

Reviewer #1: I appreciate the authors taking all comments into account. All the concerns that were previously raised by the reviewer(s) have been addressed satisfactorily—the manuscript has improved tremendously; therefore, the manuscript is acceptable in its current state.

Reviewer #2: The article is scientifically and content-wise ready and has no major problems.

There are only a few corrections in the format, references and consistency of the text that can be easily fixed

1- Go through the final version of Word once and remove duplicates.

2-In the abstract, you can add a stronger conclusion sentence.

3- Write at least one sentence about tables

7. PLOS authors have the option to publish the peer review history of their article (what does this mean? ). If published, this will include your full peer review and any attached files.

**Do you want your identity to be public for this peer review?** For information about this choice, including consent withdrawal, please see our Privacy Policy .

Reviewer #1: No

Reviewer #2: No

 Figure Resubmissions:

---

## [Editor Report · Decision Letter 3]

26 Dec 2025

Bullying among nursing university students: prevalence, characteristics, and public health implications

PGPH-D-25-01719R3

Dear Dr. Alwahaibi,

We are pleased to inform you that your manuscript 'Bullying among nursing university students: prevalence, characteristics, and public health implications' has been provisionally accepted for publication in PLOS Global Public Health.

Best regards,

Somayeh Hessam

Academic Editor